# Inertial imitation method of MMC with hybrid topology for VSC-HVDC

**Jie Wu**, **Shiyi Yin***, **Chuanjiang Li**, **Qiaozhen Zhang**

The College of Information, Mechanical and Electrical Engineering, Shanghai Normal University, Shanghai, China

* yinshiyi@shnu.edu.cn

**Data Availability Statement:** All data are included in the paper.

**Funding:** This work was supported by National Natural Science Youth Foundation of China (Grant No. 11904233). The funders had an important role in decision to publish.

## Abstract

A new virtual synchronous generator (VSG) control strategy was researched and proposed for a VSC-HVDC (High Voltage Direct Current Based on Voltage Source Converter) transmission system. It can be applied to half-bridge or full-half-bridge hybrid topology modular multi-level converter (MMC) to improve the stability and reliability of the system. First, it is proposed that the energy stored in the equivalent capacitor of MMC power module was used to imitate the rotor inertial of synchronous generator. It can buffer transient power fluctuations and synchronize autonomously with the grid. Then the impedance characteristics of the proposed control method have been deduced and analyzed. The results show that the VSG control loop mainly improves the low frequency characteristics of the converter. Secondly, the ability to suppress transient fault current is weak. So, a method, that the given values of inner current loop are calculated by grid impedance matrix, was used. A double closed loop control structure composed by a power outer loop based on VSG control and a current inner loop is obtained. The simulation results show that it can effectively improve the current control capability during the transient process for systems with a 1:2 ratio of converter capacity to grid capacity (The grid short-circuit capacity is 60MW and the MMC is 30 MW). Finally, a hybrid MMC simulation model was built based on PSCAD and the steady-state and transient fault ride-through simulations were performed. The power adjustment time of MMC under the proposed VSG control is about 1s, while the adjustment time under the conventional control strategy is greater than 4s.

## I. Introduction

Voltage source converter-based high voltage direct current transmission (VSC-HVDC) technology has emerged and developed rapidly in the past 30 years. It has been widely used in large-scale wind farms, photovoltaics and other renewable energy centralized access to the grid and asynchronous AC system interconnection. With the continuous improvement of the capacity and voltage level of the VSC-HVDC converter, the power system needs the converter to participate in the voltage and frequency regulation of the AC system. The traditional control strategy of the converter relies on the phase-locked loop (PLL) to be synchronized with the AC grid. When the AC system is weak or the new energy capacity is relatively high, this is likely to

**Competing interests:** The authors have declared that no competing interests exist.

cause system oscillations. In this regard, it is necessary to change the synchronization mechanism between the converter and the grid. Control strategies that can run autonomously need to be adopted, such as virtual synchronous generator (VSG) control [1–3], droop control [4, 5], power synchronization control [6, 7], etc. This type of control strategy synchronizes with the AC system by adjusting the grid-connected power, and can operate stably under grid-connected and islanding conditions, so there is no need to switch control structures. It is one of the research hotspots in VSC-HVDC converter control.

The VSG control method was first proposed by Lauxthal University of Technology in Germany in 2007. By simulating the complete seventh-order mathematical model of the synchronous generator, it equivalently obtains the moment of inertia and damping characteristics of the synchronous generator. Since then, the application research of virtual synchronous generator technology in converter control has been carried out, various equivalent methods of generator inertia, and improved and optimized control strategies have been proposed, for example, a control method based on frequency deviation and frequency change rate adaptively changing inertial constant and damping coefficient [8, 9], improving the equivalent damping characteristic of the system by adding a feedforward control branch to the power control branch [10, 11], and droop control used as the outer control loop to dynamically adjust the VSG operating point, enhancing system oscillation suppression capability [12, 13],and so on.

More specifically, such as the literature [14, 15], the VSG control structure is applied to the outer control loop of the doubly fed induction generator (DFIG). Its current amplitude reference signal and synchronous angular velocity of grid are calculated by VSG outer loop. In [16], a small-signal model of wind power converter based on VSG control is established. A method for optimizing the impedance characteristics of wind turbine converters is proposed based on DC capacitors, which eliminates the risk of oscillation between wind turbines and the grid. In [17], based on the conventional VSG strategy of wind turbine converters, the functional relationship between the equivalent virtual moment of inertia of the converter and its output power is established. The wind turbine state estimation function is designed to determine the level at which the wind turbine converter participates in grid frequency regulation according to its own regulation capability. However, the conventional VSG strategy does not directly control the current, and it is necessary to combine the inner current loop to suppress the transient current. In [18], the active power-virtual torque droop control and the DC voltage-frequency droop control are designed respectively. It outputs the grid synchronization signal which is used for the traditional double closed-loop control (outer power loop and inner current loop) under the synchronous rotating coordinate. In [19], the outer active power control loop is designed according to the swing equation of generator. And the state feedback method is used to improve the inner current loop of the traditional double closed-loop vector control. Then the power and current control are connected in series to form a new double closed-loop control structure. It can adaptively adjust the control parameters of the outer loop according to the fluctuation of AC frequency and power, and improve the system stability under different working conditions. In [20, 21], the converter voltage reference signal is obtained based by VSG control. And the concept of virtual synchronous impedance is proposed. It is used to connect the outer and the inner control loop to calculate the reference value of the inner current loop. It is proved that the characteristic root of the system state equation is only determined by the synchronous virtual impedance parameter, which avoids the influence of the deviation of the actual impedance parameter on the control performance.

The above research results describe in detail the idea of VSG control and its specific implementation method. VSG control has many advantages such as large stability margin for grid-connected operation, providing frequency support, and stable operation under islanding and grid-connected conditions. However, most of the research results are designed for two-level

converters. The MMC topology is more complex and has more degrees of freedom for control. For VSG control structure and inertial equivalent method, there is still room for optimization. For control structures, the modulation signal is directly output by the traditional VSG controller, without direct control of the AC current, and the fault ride-through capability is poor. Therefore, a suitable current inner loop can be constructed to improve the transient characteristics of the system. For inertial equivalent, the inertial energy in the aforementioned VSG method needs to be provided by an AC power grid, an energy storage system or a wind turbine, which requires an additional device or system to provide or consume the inertial energy. The power module capacitor of the MMC itself can be used to store energy and provide inertia. For these aspects, the main contributions of this paper include: proposing an inertial simulation method for MMC topology, designing the current inner loop to improve the transient performance, and analyzing the impedance characteristics of the converter under the proposed control strategy. The converter under the proposed control scheme can provide transient inertia for the AC system, improve the transient fault ride-through capability of the system, and provide an optimized control scheme for the weak AC system to connect to the large-capacity power electronic converter. Specifically, in the second part, the control strategy of hybrid MMC is introduced. Based on this, the method of using the energy storage in power module capacitor of the hybrid MMC itself to provide inertia for VSC-HVDC system is proposed in the third part. Then, according to the proposed virtual inertial control strategy, the impedance characteristics of the converter are derived and analyzed. A method of constructing the current inner loop is also proposed, which can be connected to the outer loop based on VSG control to improve the system's transient current control capability. And simulation verification of the proposed methods is in section four.

## II. Control mode of VSC-HVDC based on hybrid topology MMC

The VSC-HVDC system based on the full-half-bridge hybrid topology MMC can isolate permanent DC faults and pass through the transient ones. The full-bridge power module effectively improves the fault ride-through capability and the DC voltage adjustment range with less increase in converter losses. Its topology is shown in Fig 1, MMC is composed of 6 arms, all of which are composed of a certain proportion of half-bridge and full-bridge power modules mixed in series. HB is the half-bridge power module and FB is the full-bridge one. $U_{sj}$ (j = a, b, c) is three-phase AC voltage of grid side and $V_j$ is for valve side. $I_j$ is the three-phase AC current on the secondary side of the transformer. $V_{dc}$ and $I_{dc}$ are the DC voltage and DC current of the converter, respectively. And $L$ is the bridge arm reactance.

Assuming that each bridge arm is composed of $N_h$ HB power modules and $N_f$ FB ones, the ratio of $N_f$ to $N_h$ is usually greater than or equal to 50% [22, 23]. Only the FB power modules can output negative voltage, so its proportion affects the DC fault clearing ability and the reactive power output ability during transient zero dc voltage operation, etc. For hybrid topology MMC, there is usually a DC current control loop used to quickly clear DC faults and suppress short-circuit current rise. It is shown in Fig 2. $I_{dcref}$ and $I_d$ are the reference and feedback value of DC current respectively. Their errors are connected to the PI regulator, and the common mode component $V_{dc\_com}$ of the modulation signal is output. Then it is superimposed on the AC modulation signals of the 6 arms. That is, by directly changing the DC component of the bridge arm modulation signal to control the DC current to track a given value. During the transient DC fault process, the DC component of the modulation signal is set to 0 or a small negative number, which helps to clear the DC fault. The AC component of the modulation signal is generated by the current inner loop, where $I_{dref}$ and $i_d$ are the active (d-axis) current given and feedback value. $I_{qref}$ and $i_q$ are the reactive (q-axis) current reference and feedback

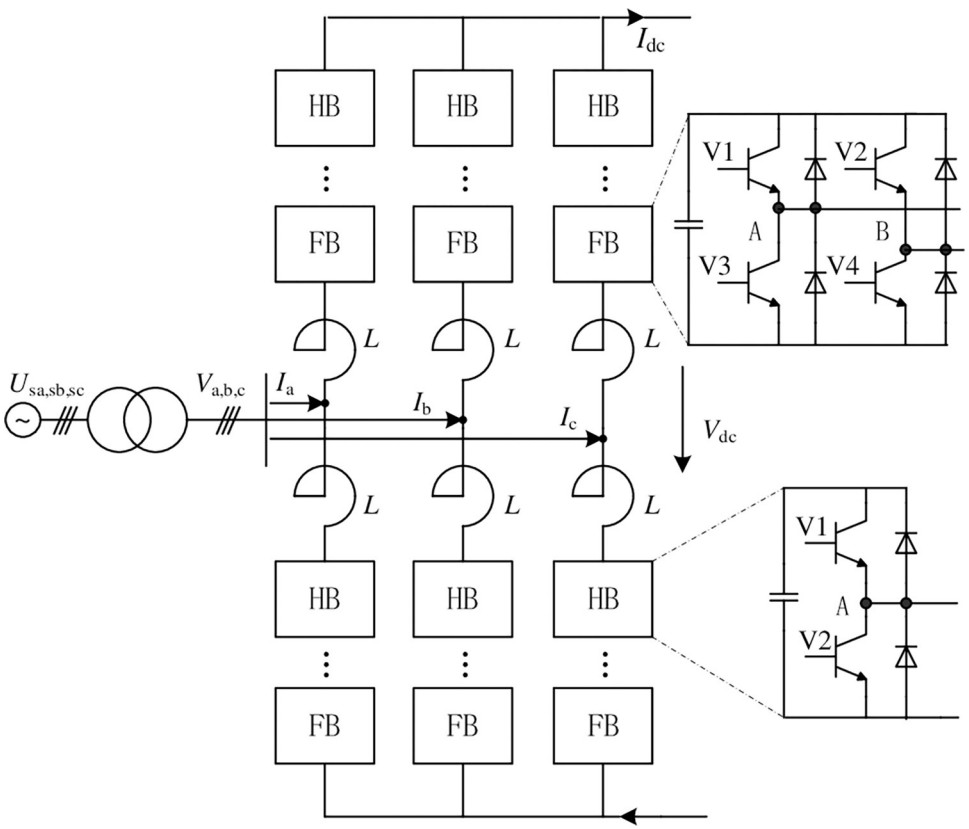

**Fig 1. Circuit of hybrid MMC topology.**

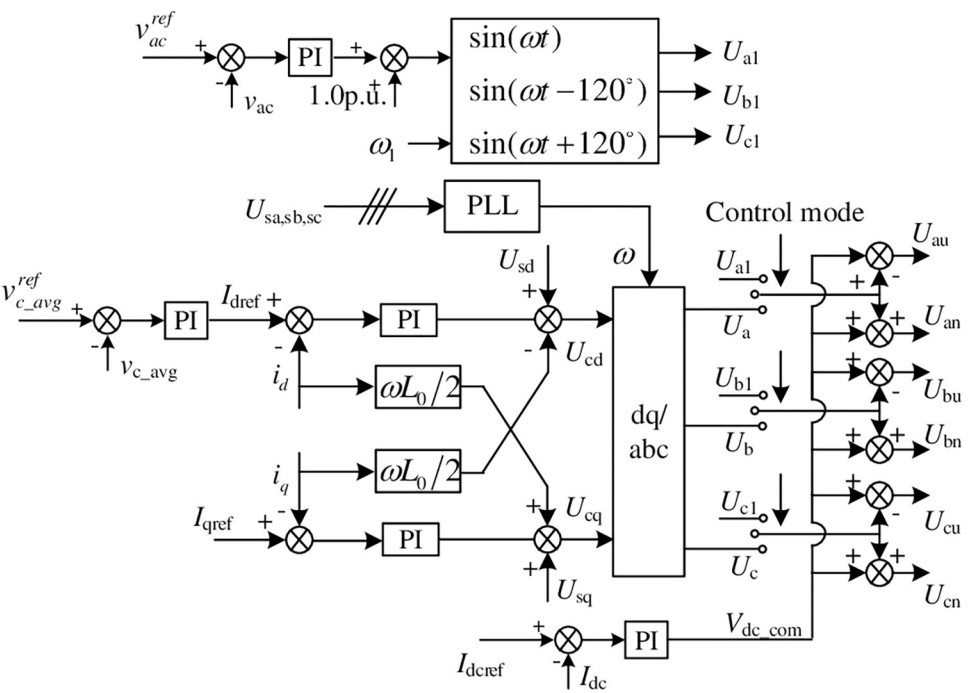

**Fig 2. Double-closed loop control structure of hybrid MMC topology.**

value respectively. PI regulators are used for dq-axis current tracking control, and their outputs are superimposed with the dq-axis feedforward component $U_{sd}$ and $U_{sq}$ of the grid voltage and the current cross-decoupling, and output $U_{cd}$ and $U_{cq}$, the dq-axis components of the AC modulation signals. The PLL is used is used to obtain the grid synchronization angular velocity $\omega$, and the $U_{cd}$ and $U_{cq}$ are transformed into modulation signals $U_j$ (j = a,b,c) of abc stationary frame. Then DC modulation signal plus and minus AC ones to get the upper and lower arm modulation signals $U_{ju}$, $U_{jn}$ (j = a, b, c), respectively. $v_{c\_avg}$ and $v_{c\_avg}^{ref}$ are the average value of all power module voltages in 6 arms and its given value. Different from the commonly used for VSC-HVDC system, the average voltage of power module capacitor instead of DC voltage is used as the control target of outer loop [22, 23]. This can maintain the AC and DC power balance during the zero DC voltage operation.

For islanding control mode, the converter supplies power to passive loads. And the three-phase AC modulation signal $U_{j1}$ (j = a,b,c) is directly calculated by the outer AC voltage amplitude loop. As shown in Fig 2, $v_{ac}^{ref}$ and $v_{ac}$ are the reference and feedback value of the AC voltage amplitude respectively, and $\omega_1$ is the rated angular frequency. the "control mode" signal is used to switch between islanding and grid connecting operation modes. In the process of control mode switching, it is difficult to synchronize the switching of the control structure and the closing of the grid-side circuit breaker, so this switching method may cause a large grid-connected inrush current.

## III. VSG control of hybrid topology MMC

### A. VSG-based average arm voltage control

As in the previous section, in the island operation mode, constant AC voltage amplitude and constant frequency control structure was used. In the grid-connected mode, the control structure of the DC capacitor voltage outer loop cascaded inner current loop is adopted, and based on the PLL to maintain synchronization with the grid. It cannot operate stably in island mode. This creates a problem that the control structure switching needs to be synchronized with the changes of islanding and grid-connected operation mode. It is difficult to synchronize them, because the mechanical structure of the AC circuit breaker will cause the closing time to be uncertain. Besides when connected to a weak AC system, the PLL output may fluctuate greatly, and the system will be unstable during startup and disturbance. In the VSG control idea, the synchronization mechanism of the AC generator is simulated. The output power of the converter is adjusted to maintain synchronization with the AC system. It can operate stably in islanding and grid-connected modes, which can avoid the switching of control structure,

A VSG control structure suitable for MMC topology is proposed in this section. The energy storage of the MMC equivalent capacitor corresponds to the inertial energy of the generator rotor. The energy storage of the power module itself is used to provide transient inertial support for the AC system. This improves system stability. And the current inner loop is designed to improve the transient current control capability of the traditional VSG. The equivalent inertia of the MMC capacitor can be obtained by comparing with rotor swing equation of the generator shown in Eq (1).

$$J\omega_m \frac{d\omega_m}{dt} + D\omega_m(\omega_m - \omega_g) = p_m - p_e \tag{1}$$

In (1), J is the moment of inertia. $P_m$ and $P_e$ are mechanical power and electromagnetic power. D is the damping coefficient. $\omega_m$ and $\omega_g$ are mechanical angular frequency of the generator rotor and the grid-side voltage, respectively. According to the principle of energy

equivalence, the equivalent capacitance $C_{eq}$ of MMC is shown in Eq (2)

$$6N\left(\frac{1}{2}C_{SM}v_c^2\right) = \frac{1}{2}C_{eq}V_{dc}^2 \tag{2}$$

Where $C_{SM}$ is the capacitance of one power module, and $N$ is the total number of full-bridge and half-bridge power modules in one arm. Considering that the capacitance of all power modules is the same, then $v_c = V_{dc} / N$. The $C_{eq}$ can be obtained as in Eq (3)

$$C_{eq} = \frac{6}{N}C_{SM} \tag{3}$$

From the power balance relationship between the AC and DC sides of the converter, Eq (4) can be obtained.

$$P_S - P_E = C_{eq}v_{c\Sigma}\frac{dv_{c\Sigma}}{dt} \tag{4}$$

$v_{c\Sigma}$ is the sum of all power module capacitor voltages. Contrast Eq (1) and Eq (4), MMC power module capacitor voltage corresponding to the angular frequency of the generator. That is, the change of $v_{c\Sigma}$ reflects the fluctuation of output power. In the steady state, the AC and DC power are balanced, and the capacitor voltage is unchanged. During the transient, the power module capacitor can buffer power fluctuations. Establish the relationship between $v_{c\Sigma}$ and $\omega$, and we can get the moment of inertia simulated by the capacitors. Here, make the linear relationship between $v_{c\Sigma}$ and the generator speed $\omega$, such as $\omega = 1/k \cdot v_{c\Sigma}$. Eq (5) can be obtained.

$$C_{eq}v_{c\Sigma}\frac{dv_{c\Sigma}}{dt} = C_{eq}k^2\omega\frac{d\omega}{dt} = J\omega\frac{d\omega}{dt} \tag{5}$$

From Eq (5), $J = C_{eq} \cdot k^2$, the proportional coefficient $k$ is determined by the allowable operating range of the MMC module voltage. The capacitance and voltage of the full-bridge and the half-bridge modules are the same, so their contribution to the inertia of the system is also the same. In order to facilitate the design of control parameters, the average value $v_{c\_avg}$ of the MMC capacitor voltage is used instead of the total capacitor voltage $v_{c\Sigma}$. The equivalent capacitance $C_{eq}$ corresponds to $J$, and the droop relationship between power and angular frequency in Eq (1) corresponds to Eq (6)

$$\omega_m = \omega_n + \frac{1}{k}\left(v_{c\_avg} - v_{c\_avg}^{ref}\right) \tag{6}$$

In Eq (6), $\omega_n$ and $\omega_m$ are the angular frequency of grid and converter output voltage. There is a drooping relationship between $\omega_m$ and $v_{c\_avg}$. Based on the above, the converter control structure is proposed as shown in Fig 3. $k_{vc}$ and $k_v$ are the drooping coefficients of the capacitor voltage outer loop and reactive power outer loop, respectively. The phase $\theta$ and amplitude $E_{ref}$ of modulation signal transformed by polar coordinates to obtain the dq axis component of the modulation signal $E_{dref}$ and $E_{qref}$. Then the modulation signal is calculated by the inner current loop, which is transformed from the synchronous rotating coordinate to the stationary coordinate system as $u_{abc}^{ac}$ in Fig 3.

The converter outputs active power to the AC grid as the positive direction. When the average value of the module voltage is less than the reference value, it indicates that the output power to the AC grid has increased. The power module capacitor energy storage is converted into transient power support of grid, and the capacitor voltage in the steady state will be slightly lower than the rated value. The phase difference between the AC voltage vector of the converter and the grid voltage vector will increase, stabilizing at the new operating point. After

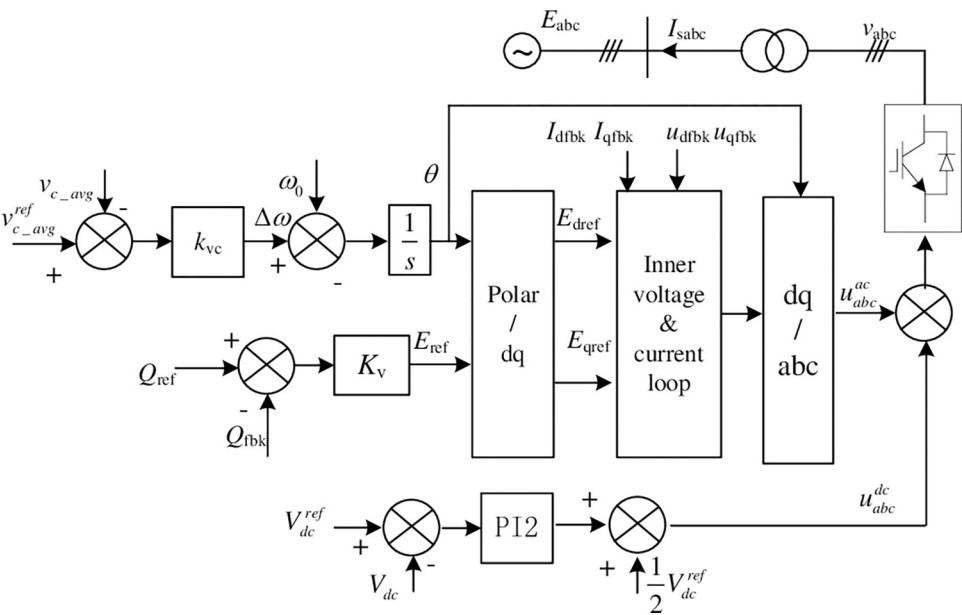

**Fig 3. Constant DC voltage control structure based on VSG strategy.**

that, other converters connected in parallel on the DC bus will inject power into this station from the DC side, and the voltage of the sub-modules in this station gradually recovers, which can continuously provide greater inertial support power to the AC system, which is equivalent to increasing the system's moment of inertia.

In Fig 3, droop control is also used for the reactive power control loop, which adjusts the AC output voltage of the converter according to the reactive power deviation between the reference and actual value. This is similar to the principle of reactive power control of generators. PI regulator is used for DC voltage control loop, and the output is the common mode component of the upper and lower arm modulation signals. When the average voltage of the power module capacitors deviates from the rated value, the DC bus voltage is kept constant by adjusting the number of modules with on-state in the upper and lower arms.

## B. Current control and system impedance characteristics

From the main circuit in Fig 1 and Kirchhoff's law, the AC side equation of the converter can be obtained as:

$$\begin{cases} L\dfrac{di_d}{dt} = U_{sd} - U_{cd} + \omega L i_q - R i_d \\ L\dfrac{di_q}{dt} = U_{sq} - U_{cq} - \omega L i_d - R i_q \end{cases} \tag{7}$$

$R$ is the equivalent resistance of the AC circuit. The DC side model of the converter is obtained as

$$C\frac{dv_{c\_avg}}{dt} = \frac{3(U_{cd}i_d + U_{cq}i_q)}{2V_{dc}} - I_{dc} \tag{8}$$

Eq (7) and Eq (8) are the mathematical model of the main circuit of the MMC. Considering that the control of inner loop contains current dq-axis cross-decoupling links, the coupling of the dq-axis current in the main loop can be removed, and Eq (7)–(8) can be linearized as

shown in Eq (9).

$$\begin{cases} \begin{bmatrix} \Delta i_d \\ \Delta i_q \end{bmatrix} = (\mathbf{I} - \mathbf{A_1})^{-1} \cdot \mathbf{B_1} \cdot \mathbf{u_1} + (\mathbf{I} - \mathbf{A_1})^{-1} \cdot \mathbf{B_2} \cdot \mathbf{u_2} \\ \Delta v_{c\_avg} = (\mathbf{B_3} \cdot (\mathbf{I} - \mathbf{A_1})^{-1} \cdot \mathbf{B_1} + \mathbf{B_4}) \cdot \mathbf{u_1} \\ \qquad\qquad + \mathbf{B_3} \cdot (\mathbf{I} - \mathbf{A_1})^{-1} \cdot \mathbf{B_2} \cdot \mathbf{u_2} \end{cases} \tag{9}$$

In the equation:

$$\mathbf{A_1} = \begin{bmatrix} 0 & \dfrac{\omega_0}{s} \\ -\dfrac{\omega_0}{s} & 0 \end{bmatrix}, \mathbf{B_1} = \begin{bmatrix} -\dfrac{1}{sL} & 0 \\ 0 & -\dfrac{1}{sL} \end{bmatrix}, \mathbf{B_2} = \begin{bmatrix} -\dfrac{E \cdot \sin\theta_0}{sL} \\ -\dfrac{E \cdot \cos\theta_0}{sL} \end{bmatrix}, \mathbf{B_3} = \begin{bmatrix} \dfrac{3u_{d0}}{2CV_{dc}s} \\ \dfrac{3u_{q0}}{2CV_{dc}s} \end{bmatrix}',$$

$$\mathbf{B_4} = \begin{bmatrix} \dfrac{3i_{d0}}{2CV_{dc}s} \\ \dfrac{3i_{q0}}{2CV_{dc}s} \end{bmatrix}'$$ .It is a two-input two-output linear system. The two inputs include the dq

axis modulation signal $\mathbf{u_1} = [\Delta u_d \quad \Delta u_q]'$., and the phase of AC voltage vector of converter side, $\mathbf{u_2} = \Delta\theta$. The two outputs include the dq axis current of the converter, $[\Delta i_d \quad \Delta i_q]'$ a the average of the sub-module capacitor voltage $\Delta v_{c\_avg}$.

The state equations of the system control loop can be obtained from Fig 3. The active power control loop is as in Eq (10).

$$\frac{d\theta}{dt} = \omega_n + k_{vc}(v_{c\_avg} - v_{c\_avg}^{vef}) \tag{10}$$

The reactive control loop is as in Eq (11).

$$E_{ref} = (k_Q + k_{Qi}/s)(Q_{re} - Q_{fbk}) \tag{11}$$

PI regulators are used for inner AC voltage and current control loop. Linearize the control loop shown in Eq (12).

$$\begin{cases} \Delta\theta = \dfrac{k_{vc}\Delta v_{c\_avg}}{S} \\ \Delta u_{dref} = k_v \Delta Q \\ \Delta i_{dref} = PI_1 \cdot (\Delta u_{dref} - \Delta u_d) \\ \Delta i_{qref} = PI_2 \cdot (0 - \Delta u_q) \\ \Delta u_d = PI_3 \cdot (\Delta i_{dref} - \Delta i_d) \\ \Delta u_q = PI_3 \cdot (\Delta i_{qref} - \Delta i_q) \end{cases} \tag{12}$$

The system small signal transfer function structure is obtained from the main loop and control loop linearization Eqs (9) and (12) shown in the Fig 4.

In the figure, the reactive power small signal feedback ($\Delta Q_{fbk}$) is shown as Eq (13).

$$\Delta Q_{fbk} = E \cdot \left( \begin{bmatrix} \sin\theta_0 & \cos\theta_0 \end{bmatrix} \begin{bmatrix} \Delta i_d \\ \Delta i_q \end{bmatrix} + (i_{d0}\cos\theta_0 - i_{q0}\sin\theta_0) \cdot \Delta\theta \right) \tag{13}$$

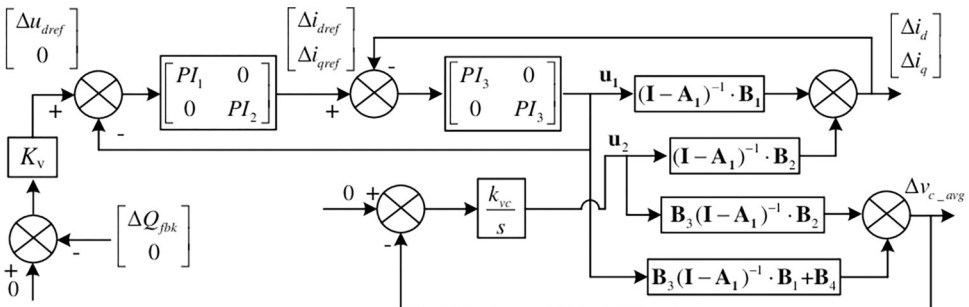

**Fig 4. Converter small signal transfer function structure of constant DC voltage station.**

Each element in the coefficient matrix can be obtained from the steady-state operating point of the system, as shown in Eq (14).

$$
\begin{cases}
i_{d0} = \dfrac{P_0}{E} \cdot \sqrt{\dfrac{2}{3}} \\[2mm]
i_{q0} = \dfrac{Q_0}{E} \cdot \sqrt{\dfrac{2}{3}} \\[2mm]
\theta_0 = \arcsin\left(\dfrac{\omega_0 L_{eq} i_{d0}}{E}\right) \\[2mm]
0 = E\cos\theta_0 - u_{d0} + \omega_0 L_{eq} i_{q0} \\[2mm]
0 = E\sin\theta_0 - u_{q0} - \omega_0 L_{eq} i_{d0}
\end{cases}
\tag{14}
$$

From the small signal transfer function structure of the system in Fig 4, the relationship between $\begin{bmatrix} \Delta u_d & \Delta u_q \end{bmatrix}'$ and $\begin{bmatrix} \Delta i_d & \Delta i_q \end{bmatrix}'$ can be obtained as Eq (15), which can be used for the system impedance calculation.

$$
\begin{bmatrix} \Delta u_d \\ \Delta u_q \end{bmatrix} = \begin{bmatrix} PI_3 & 0 \\ 0 & PI_3 \end{bmatrix} \cdot \left( \begin{bmatrix} PI_1 & 0 \\ 0 & PI_2 \end{bmatrix} \cdot \left( \left( \begin{bmatrix} \Delta u_{dref} \\ 0 \end{bmatrix} - \begin{bmatrix} \Delta u_d \\ \Delta u_q \end{bmatrix} \right) - \begin{bmatrix} \Delta i_d \\ \Delta i_q \end{bmatrix} \right) \right)
\tag{15}
$$

In Eq (15), $\begin{bmatrix} \Delta u_{dref} & 0 \end{bmatrix}'$ is related to the converter input variables $\mathbf{u_1}$ and $\mathbf{u_2}$, so the relationship between $\mathbf{u_1}$ and $\mathbf{u_2}$ is first derived from Fig 4. It is as Eq (16).

$$
\mathbf{u_2} = \frac{[B_3(I - A_1)B_1 + B_4]\frac{k_v}{s}}{\frac{s}{k_{vc}} - [B_3(I - A_1)B_2]} \mathbf{u_1}
\tag{16}
$$

From Eq (12) and Eq (16)

$$\begin{bmatrix} \Delta u_{dref} \\ 0 \end{bmatrix} = Ek_v \left( \begin{bmatrix} \sin\theta_0 & \cos\theta_0 \\ 0 & 0 \end{bmatrix} \begin{bmatrix} \Delta i_d \\ \Delta i_q \end{bmatrix} + \frac{i_0 \dfrac{k_v}{s}}{\dfrac{s}{k_{vc}} - b_0} \begin{bmatrix} b_{11} & b_{12} \\ 0 & 0 \end{bmatrix} \begin{bmatrix} \Delta u_d \\ \Delta u_q \end{bmatrix} \right) \quad (17)$$

where

$i_0 = i_{d0}\cos\theta_0 - i_{q0}\sin\theta_0, \ b_0 = [B_3(I - A_1)B_2],$
$[b_{11} \quad b_{12}] = [B_3(I - A_1)B_2].$

Bringing Eq (17) into Eq (15), we can get the system impedance model under the proposed control strategy. Take the impedance $z_{dd}$ between $\Delta u_d$ and $\Delta i_d$ as an example, the impedance characteristic of $z_{dd}$ shown in Fig 5.

According to the theoretical derivation and impedance scanning results, it can be seen that the system has an inductive characteristic in the high frequency region (>300Hz), which is

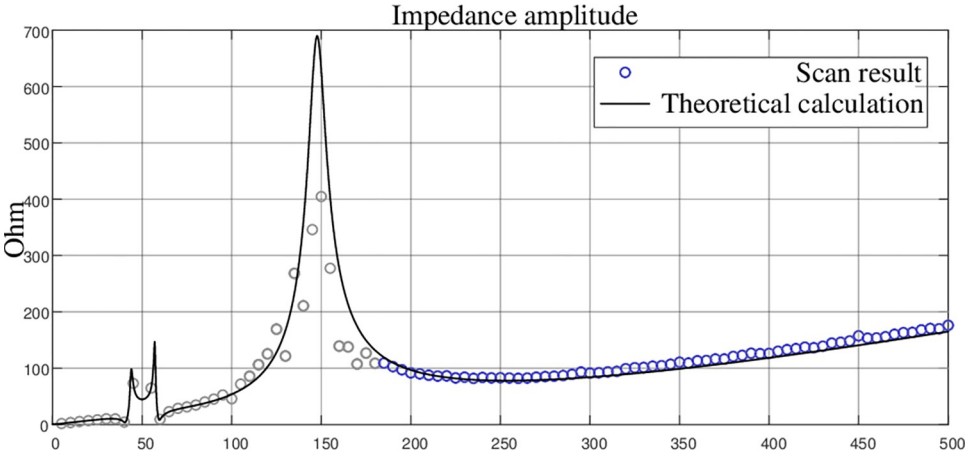

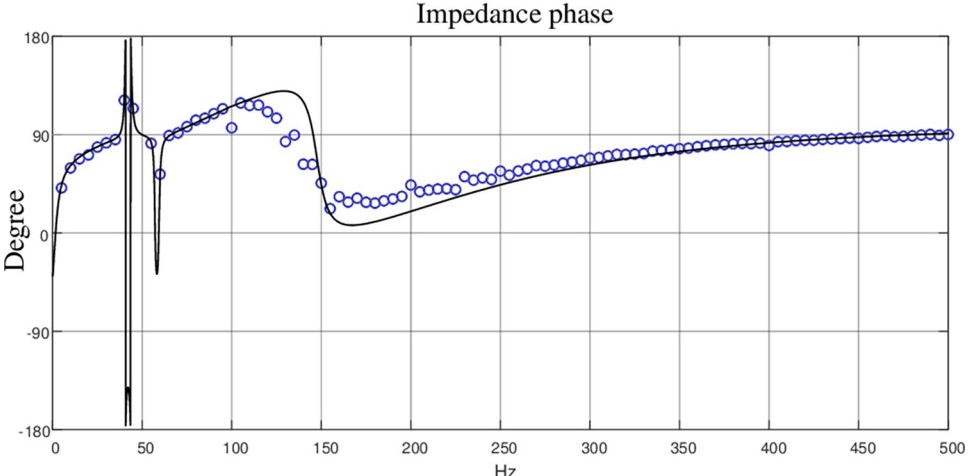

**Fig 5. AC port Impedance characteristics of constant DC voltage station.**

similar to the traditional double closed-loop control strategy. The proposed virtual inertia control method mainly affects the impedance of the system in the low-frequency region.

## C. VSG control of active power

For the converter station with active power as the control target in VSC-HVDC system, the control loop of the common mode component of the modulation signal needs to be changed compared with the constant DC voltage control station. It is shown in Fig 6, where $P_{ref}$ and $P_{fbk}$ are the reference and feedback value of active power respectively. The output of the active power regulator PI1 is the DC current reference $I_{dc\_ref}$, and $I_{dc}$ is the feedback. The output of the DC current regulator PI2 is the common mode component of the modulation signal. It changes the DC side transmission power of the converter by adjusting the DC output voltage, so that the active power can track the reference value. The voltage and reactive power control structure of the power module in this figure is the same as that in Fig 3.

For the above control strategies of the two ends, in Figs 3 and 6, it is droop control characteristic between $V_{c\_avg}$ and $\omega$ in the outer control loop, and the energy during the transient process is buffered by changing the capacitor voltage. The capacitors provide inertia. The common mode component of the modulated signal can be used to control the DC voltage and active power to track the reference signals.

For the constant power control converter, the conventional VSG control loop can also be used. As shown in Fig 7, a first-order inertial link between the $P$ and $\theta$ is used for inertial imitation. It can be represented by the Eq (18).

$$\begin{cases} \omega - \omega_0 = \dfrac{P_m - P_e}{2Hs + D} \\ \theta = \dfrac{\omega}{s} \end{cases} \tag{18}$$

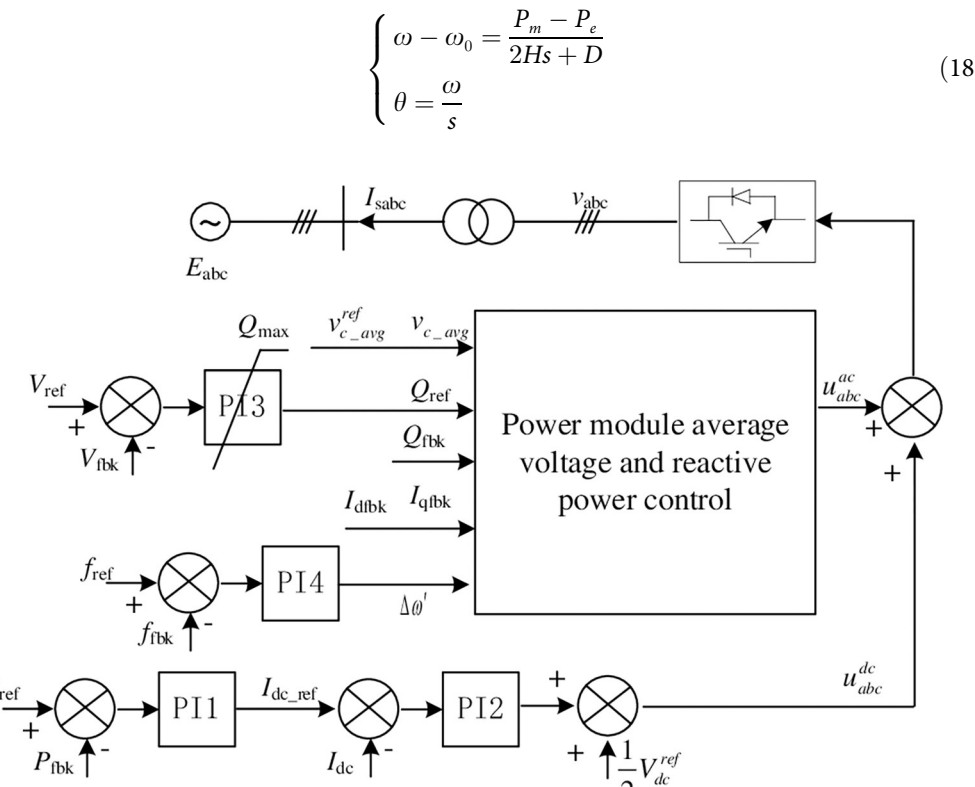

**Fig 6. Constant active power control structure based on VSG strategy.**

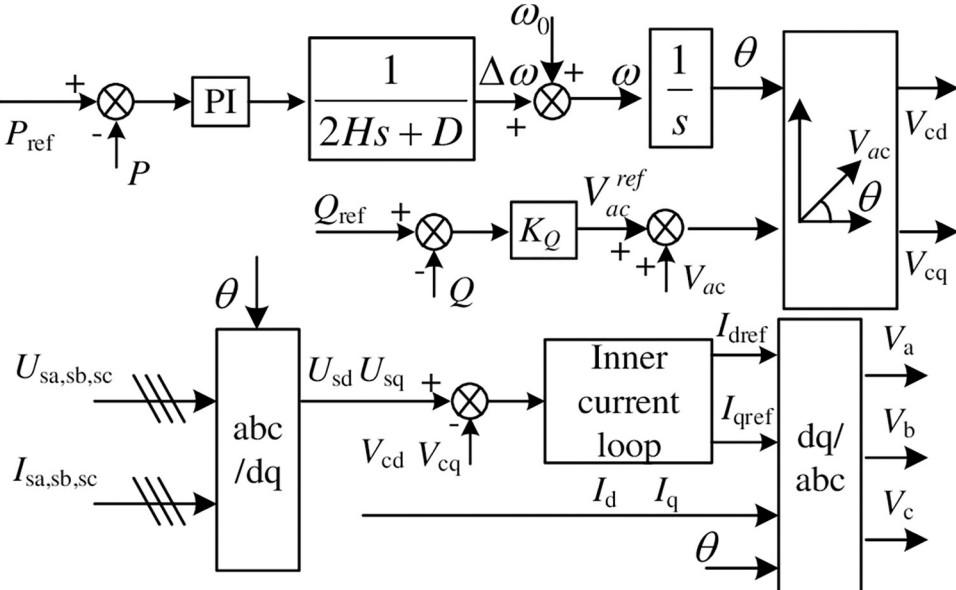

**Fig 7. Constant active power control structure based on traditional VSG strategy.**

In Eq (18), $s$ is differential operator. Based on swing equation of synchronous machine, the grid synchronization angular frequency $\omega$ can be obtained. $P_m$ and $P_e$ are mechanical power and electromagnetic power. $D$ is the damping coefficient. H is the inertia time constant. The inner current loop is the same as Fig 3. For the control structure of Fig 7, the inertial support energy needs to be supplied by the active power input from the DC side, not by the MMC capacitor energy storage. Therefore, it requires additional energy storage or power supply on the DC side. This is the main difference from the control structure of the outer VSG control loop proposed in this paper.

## IV. Simulation and analysis

In order to verify the proposed VSG control strategy, a VSC-HVDC system based on full-half-bridge hybrid MMC is built based on PSCAD. One terminal takes the active power or AC voltage and frequency as the control target. The control structure is shown in Fig 6. The other one takes the DC voltage as the control target and the control structure is shown in Fig 3. The control parameters of the VSG outer loop are determined by the impedance characteristics of the small signal model, and the current inner loop controller parameters are mainly adjusted through simulation, so that the adjustment speed of the current inner loop is significantly higher than that of the outer loop. The conventional circulation suppression method is adopted for MMC station based on double-frequency rotating coordinate [24]. And The power module voltage sequencing method is used to maintain voltage balance during both steady state and transient state [25]. The main circuit structure is shown in Fig 8. The AC sides of the converters MMC1 and MMC2 at both ends are connected to the transformers $T_1$ and $T_2$ respectively. The equivalent inductance $L_1$ and $L_2$ of the grid is 5mH. The grid voltage $U_{s1}$ and $U_{s2}$ is 10 kV, so the short circuit capacity of the system is about 60 MW, MMC connected with weak AC system. The MMC1 can smoothly switch between islanding and grid-connected modes through circuit breakers CB1 and CB2. There are 10 power modules connected in series in one arm, including 5 half-bridge modules and 5 full-bridge ones. Power module rated voltage is 2kV and the DC voltage of VSC-HVDC system is 20 kV. Its rated power is 30 MW. The

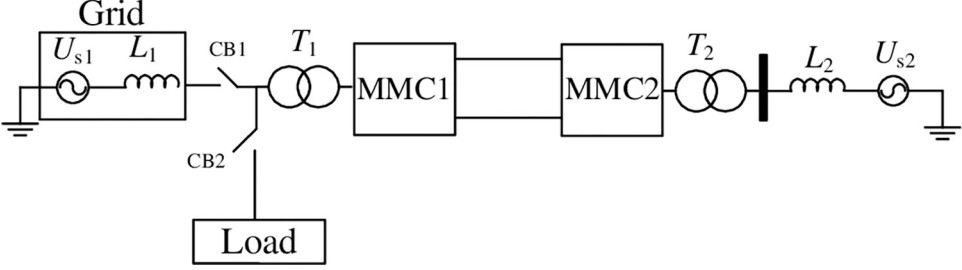

**Fig 8. Simulation structure based on VSG strategy.**

parameters in the simulation model are obtained by reducing the DC voltage level according to the VSC-HVDC project for reducing the amount of calculation [26], and keep the DC current at the same level.

System parameters are normalized with peaks of AC voltage and AC current in the simulation. Active and reactive power is normalized by system rated capacity. And adjusted by simulation, $J$ = 0.0471 p.u., $D$ = 1.1p.u. The AC side of MMC1 is connected to a passive load, with island control mode. The MMC2 is connected to the grid, and the DC voltage is controlled to the rated value. At t = 3s, the grid-connected operation signal is sent to MMC1, and the controller detects the phase difference between the output voltage of the MMC1 and the grid voltage. When the phase difference meets the requirements, the pre-synchronization process ends, and the grid-side circuit breaker CB1 is closed. The simulation results are shown in Fig 9. In island mode, the given frequency $F_{ref}$ is higher than the rated frequency of the grid, which is 1.01p.u. shown in Fig 9(A). Due to the deviation between $F_{ref}$ and the actual frequency of the power grid, the active power changes after the grid is connected. The $P$-$\omega$ link in Fig 3 quickly adjusts the angular frequency of the MMC output voltage to achieve self-synchronization with the power grid. the $P_{ref}$ and $P$ in Fig 9(B) are the reference and feedback of the $P$-$\omega$ link, and its output is the angular frequency adjustment value $\Delta\omega$ shown in Fig 9(C). this makes the MMC to track the grid frequency autonomously. As shown in the simulation results, although there is large static-state error during the grid connection process, the MMC can maintain synchronization with the power grid without switching the control structure, which can simplify the switching process and maintain the stable operation of the system. After the grid-connected process is over, conventional secondary regulation and other methods can be used to adjust the frequency reference signal to be the same as the actual frequency of the grid, and the active power will also track its reference.

In order to verify the transient power support performance of the proposed control method and the stability of access to the weak grid, at $t$ = 4s active power steps change from 0MW to 30 MW. The simulation results are shown in Fig 10. The variable label "1" in the figure is the output waveform of the conventional double closed loop control method (Fig 2) for hybrid topology, and the label "2" is the output waveform based on VSG strategy.

Fig 10(A) and 10(B) are the waveforms of DC voltage and DC current, respectively. As shown in the figures, with conventional control strategy, there are large oscillations when the converter is unlocked and during the power step rising. The oscillation attenuation is slow, so the system characteristic is of weak damping. Fig 10(C) is the step response of active power. It can be seen from the control method shown in Fig 6 that the active power regulator will increase the common mode component of the modulation signal. This increases the on-state power modules, temporarily release the energy of the power module, and reduce the impact on the power grid. Then the average capacitor voltage control loop $v_{c\_avg}$–$\Delta\omega$ adjusts the output angular frequency. That is, by adjusting the phase of the AC output voltage to increase the

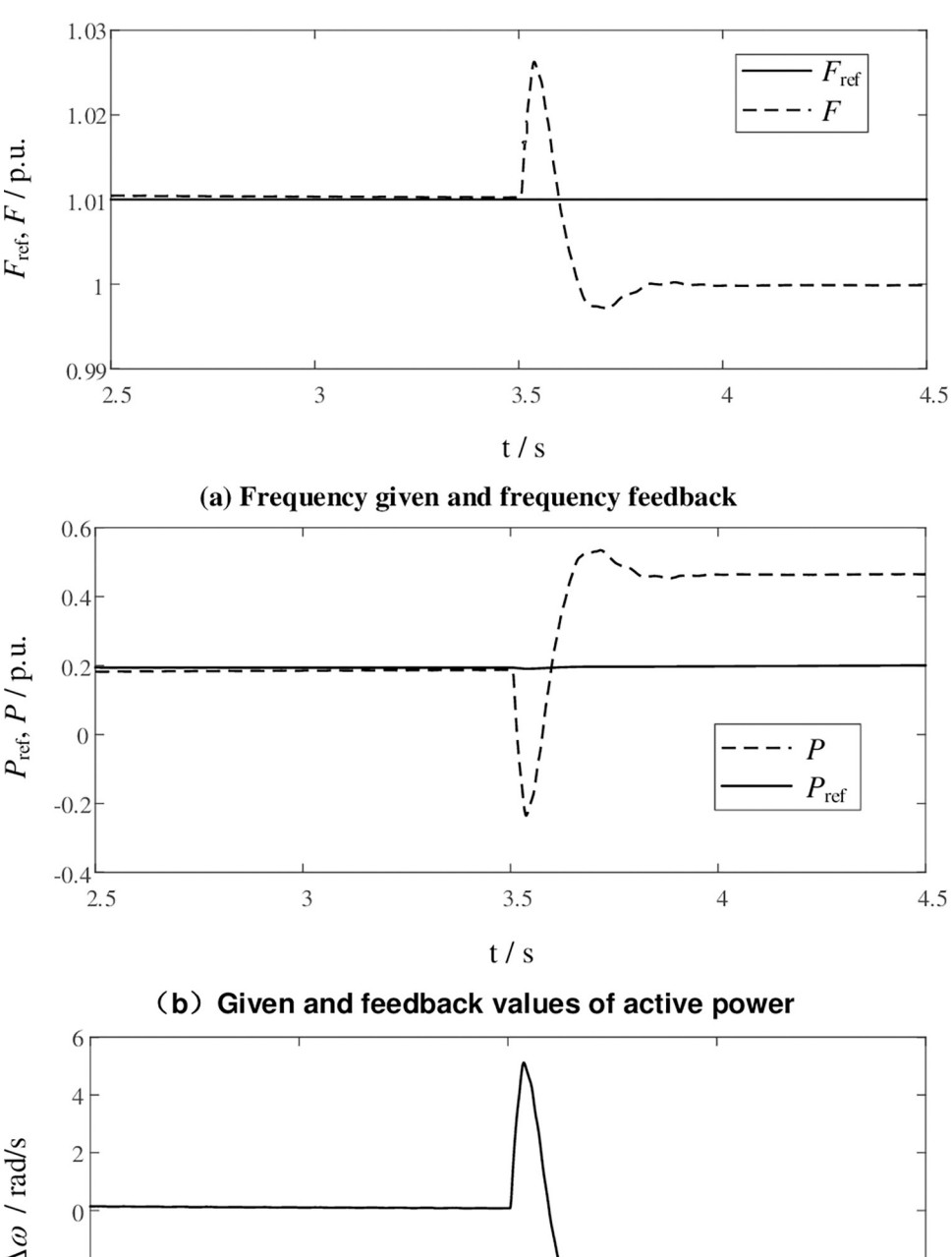

(a) Frequency given and frequency feedback

（b）Given and feedback values of active power

(c) angular frequency adjustment value

**Fig 9. Switch form island to grid connection of MMC1.**

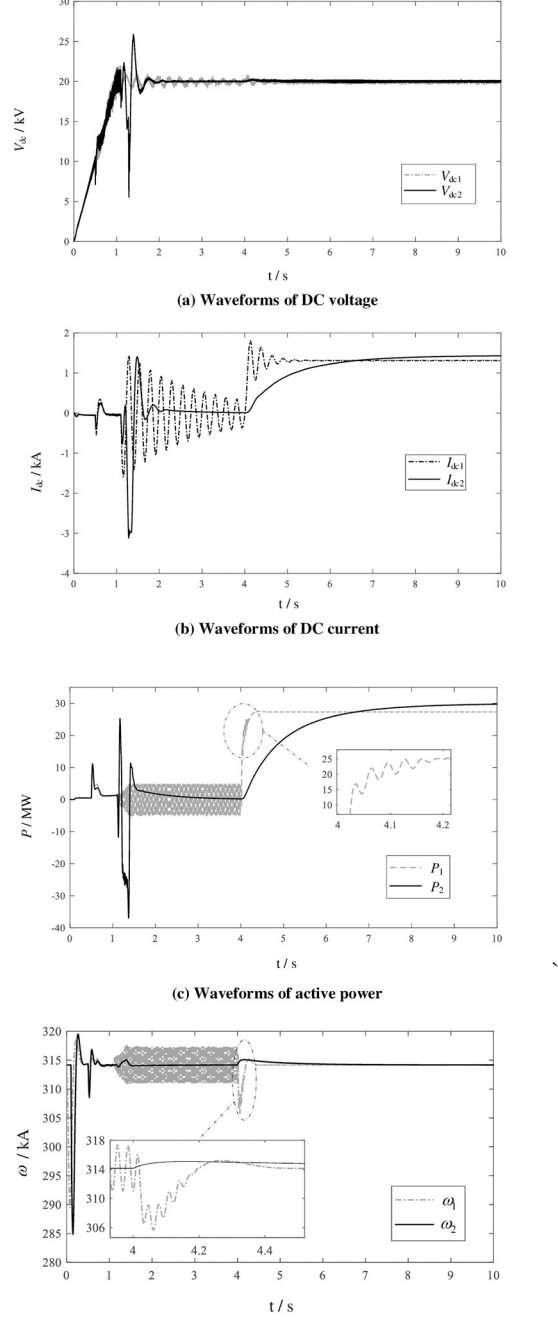

(a) Waveforms of DC voltage

(b) Waveforms of DC current

(c) Waveforms of active power

(d) Grid synchronization angular velocity calculated by PLL and active power controller based on VSG

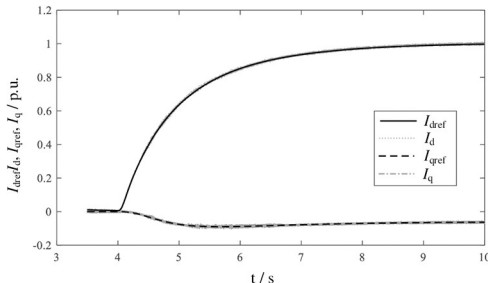

(e) Tracking the reference values of dq axis inner current loop

**Fig 10. Active power step response by conventional dual closed loop control and VSG control used for hybrid topology.**

input active power. Fig 10(D) shows the PLL output $\omega_1$ of the conventional control strategy and the grid synchronization angular frequency $\omega_2$ calculated by the $v_{c\_avg}$–$\Delta\omega$ control loop. It can be seen that the PLL output fluctuations during the unlocking and step response, indicating that if the weak AC system connected, the PLL characteristics deteriorate during the transient process, causing the system oscillation. Fig 10(E) shows the dq-axis current reference and feedback values during the step process based on VSG control strategy, and both d- and q-axis currents can track the given values during steady-state and step processes.

Hybrid topology MMC can suppress DC short-circuit current. For verification, DC bipolar short-circuit fault is set at t = 8s. The fault is cleared after 100 ms, and the simulation results are shown in Fig 11. And Fig 11(A) shows the DC voltage and current waveform during the DC fault. Fig 11(B) shows the dq-axis current reference and feedback, and there is a small fluctuation of feedback currents during the short-circuit process. Fig 11(C) also shows the dq-axis current reference and feedback based on reference [16] for comparison with the method in this paper. In [16], the VSG outer loop and the current inner loop structure are also used. But its inertial energy is provided by the wind turbine instead of the converter's own capacitors. Although the wind turbine can provide greater inertial energy, in the process of transient DC fault, in order to keep the capacitor voltage constant, its current inner loop given value and feedback value fluctuate greatly, resulting in overcurrent blocking of the converter. Fault ride-through failure. In the method proposed in this paper, the VSG outer loop is constructed based on the capacitor energy storage of the power module. During the transient fault process, the given value of the power module voltage will be increased, which is the same as the change trend of the feedback value, thereby reducing the fluctuation of the current inner loop. Fig 11 (D) shows the three-phase current ($I_{an}$, $I_{bn}$, $I_{cn}$) of lower arms of the converter. During DC fault, the output of DC current control module rapidly reduces, which is the DC component of the modulation signals. This can make the DC voltage close to 0, suppress the DC short-circuit current from rising, and realize the transient DC fault clearing.

## V. Conclusion

First of all, the control method of MMC can autonomously keep synchronous with the grid. It can solve the problem of strong coupling between the control structure and the main circuit operation mode in the traditional PLL-based control strategy, and simplify the switching process between islanding and grid connection of VSC-HVDC.

Secondly, based on the energy storage of MMC power modules, VSC-HVDC system can imitate synchronous generator characteristics. Capacitor average voltage of arms can be controlled by adjusting the angular frequency of voltage vector of converter side. It can buffer active power changes during transient process and improve system stability. When the converter valve is connected to the weak AC system, it can eliminate the system oscillation caused by the PLL during transient and effectively improve system stability.

Finally, through the analysis of the impedance characteristics of the VSG, it is shown that the outer loop of the VSG mainly affects the low-frequency characteristics of the system, and the high-frequency characteristics are still dominated by the current inner loop. The proposed control structure is of positive impedance at low frequency and can buffer the influence of the transient process on the AC system. In summary, the proposed VSG control technology applied to hybrid topology MMC can effectively improve the system stability and reliability of VSC-HVDC.

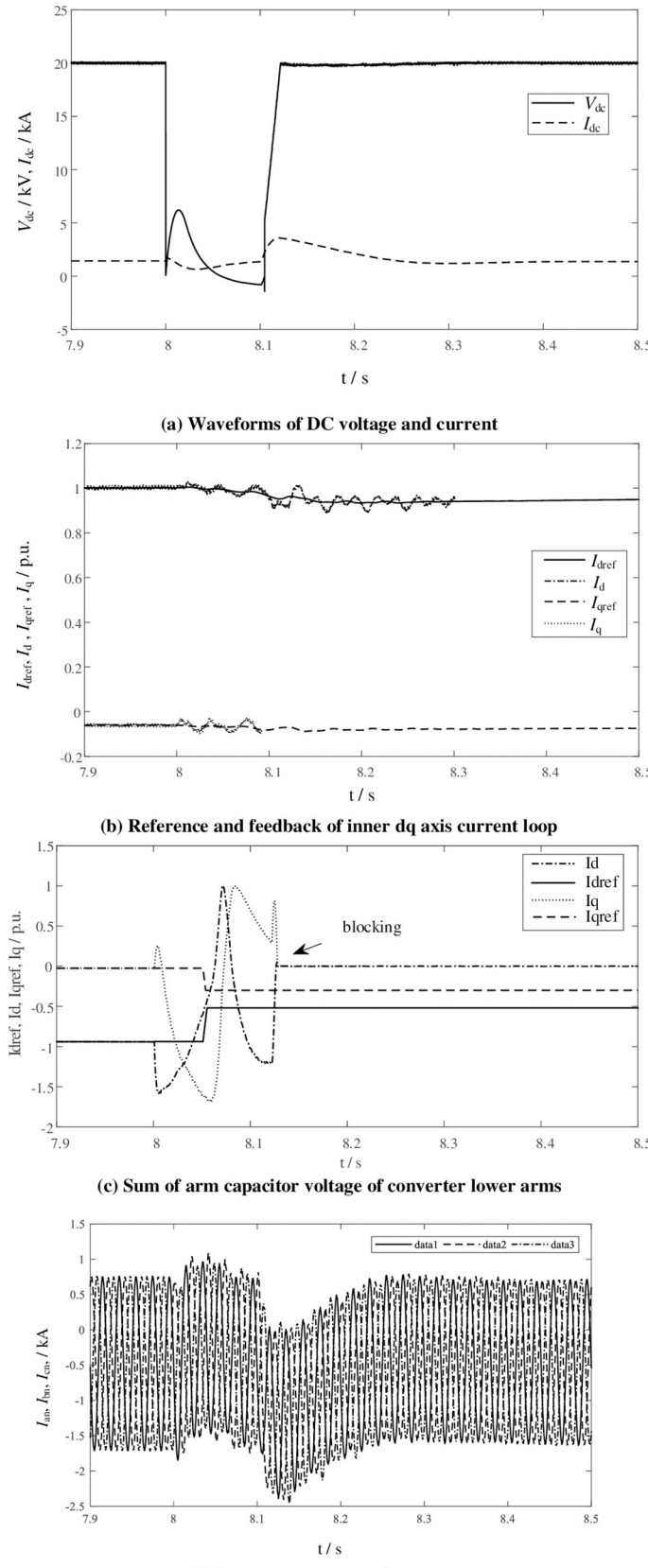

**(a) Waveforms of DC voltage and current**

**(b) Reference and feedback of inner dq axis current loop**

**(c) Sum of arm capacitor voltage of converter lower arms**

**(d) Lower arm current of converter**

**Fig 11. DC short-circuit fault simulation waveform.**

## Author Contributions

**Methodology:** Jie Wu.

**Project administration:** Chuanjiang Li.

**Supervision:** Chuanjiang Li.

**Validation:** Shiyi Yin, Qiaozhen Zhang.

**Writing – original draft:** Jie Wu.

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
