## [Decision Letter · Decision Letter 0]

17 May 2022

PONE-D-22-05125Inertial Imitation Method of MMC with Hybrid Topology for VSC-HVDCPLOS ONE

Dear Dr. wu,

Thank you for submitting your manuscript to PLOS ONE. After careful consideration, we feel that it has merit but does not fully meet PLOS ONE’s publication criteria as it currently stands. Therefore, we invite you to submit a revised version of the manuscript that addresses the points raised during the review process.

We look forward to receiving your revised manuscript.

Kind regards,

Yogendra Arya

Academic Editor

PLOS ONE

Journal Requirements:

2. Please note that PLOS ONE has specific guidelines on code sharing for submissions in which author-generated code underpins the findings in the manuscript. In these cases, all author-generated code must be made available without restrictions upon publication of the work. Please review our guidelines at https://journals.plos.org/plosone/s/materials-and-software-sharing#loc-sharing-code and ensure that your code is shared in a way that follows best practice and facilitates reproducibility and reuse

Reviewers' comments:

Reviewer's Responses to Questions

**Comments to the Author**

1. Is the manuscript technically sound, and do the data support the conclusions?

Reviewer #1: Yes

Reviewer #2: Partly

2. Has the statistical analysis been performed appropriately and rigorously? 

Reviewer #1: Yes

Reviewer #2: N/A

3. Have the authors made all data underlying the findings in their manuscript fully available?

Reviewer #1: Yes

Reviewer #2: Yes

4. Is the manuscript presented in an intelligible fashion and written in standard English?

Reviewer #1: Yes

Reviewer #2: No

5. Review Comments to the Author

Reviewer #1: 1.The authors should make clear the contribution of the manuscript.

2.How can the authors get the parameters for the case study, because all the parameters in the case are different from the actual project.

3.Some information is missing in 【13】,【24】,etc, and no CB01 in Fig.8.

Reviewer #2: This Paper proposes "Inertial Imitation Method of MMC with Hybrid Topology for VSC-HVDC"

Please consider the following comments.

1. The obtained results have not been discussed in the abstract section.

2. There are many virtual inertia emulation techniques in the literature. The authors should compare one of those recent techniques with the proposed techniques.

3. Please find the gap first in the introduction section by discussing several recent studies in VSG. Then, write your contributions.

3. "For VSG control structure and inertial equivalent method, there is still room for optimization." Is there any optimization the proposed method?

5. Why dq frame control strategy is used?

6. How the controller parameters have been tuned? If the tuning has been done based on the frequency response analysis, a complete small-signal model is required?

7. How the capacitor voltage balancing is achieved during worst case scenario?

8. The quality of the figures of the simulations results should be improved.

9. Is the proposed control system a multi input multi output (MIMO)? Please explain.

6. PLOS authors have the option to publish the peer review history of their article (what does this mean?). If published, this will include your full peer review and any attached files.

Reviewer #1: No

Reviewer #2: No

---

## [Author Response · Author response to Decision Letter 0]

23 Jul 2022

All the responses are in the document "Response to Reviewers", it has been uploaded, the main content is as follows：

Reviewer #1:

(1) The authors should make clear the contribution of the manuscript.

In the first part, the contributions of the paper are explicitly presented, supplemented as follows: 

“For these aspects, the main contributions of this paper include: proposing an inertial simulation method for MMC topology, designing the current inner loop to improve the transient performance, and analyzing the impedance characteristics of the converter under the proposed control strategy. The converter under the proposed control scheme can provide transient inertia for the AC system, improve the transient fault ride-through capability of the system, and provide an optimized control scheme for the weak AC system to connect to the large-capacity power electronic converter.” 

And the original related narrative content " And the influence of the new control strategy on the impedance characteristics of the converter also needs to be further analyzed. For these, in this paper, a VSG control method is proposed for MMC topology, which can increase the inertia of the converter and improve the stability of the system." is deleted

(2) How can the authors get the parameters for the case study, because all the parameters in the case are different from the actual project.

In Section 4, the description of simulation model parameter settings has been supplemented, and corresponding references have been added. It is “The parameters in the simulation model are obtained by reducing the DC voltage level according to the VSC-HVDC project for reducing the amount of calculation, and keep the DC current at the same level. In the reference “Analysis of Resonance Between a VSC-HVDC Converter and the AC Grid” published in IEEE Transactions on Power Electronics”, there are several VSC-HVDC projects with a DC current of about 1500 A, for example ±320 kV/1000 MW Xiamen project, the ±350 kV/1000 MW Luxi back-to-back project. Therefore, the rated DC current is set to 1500 A in the simulation model in this paper, and the DC voltage and capacity are proportionally reduced.

(3) Some information is missing in [13],[24], etc, and no CB01 in Fig.8.

The information of [13], [24] has be supplemented. And the CB01 was changed to CB1 in Fig.8.

Reviewer #2:

(1) The obtained results have not been discussed in the abstract section.

The results were added in abstract as “The results show that the VSG control loop mainly improves the low frequency characteristics of the converter”, and “the steady-state and transient fault ride-through simulations were performed. the results show that the proposed method is effectiveness.”

(2) There are many virtual inertia emulation techniques in the literature. The authors should compare one of those recent techniques with the proposed techniques.

The VSG method in reference [16] is used to compare with ours, which is published in year 2021. And the Fig. 11(c) is replaced.

The explanation about Fig. 11(c) is supplemented as follows: 

Fig. 11(c) also shows the dq-axis current reference and feedback based on reference [16] for comparison with the method in this paper. In [16], the VSG outer loop and the current inner loop structure are also used. But its inertial energy is provided by the wind turbine instead of the converter's own capacitors. Although the wind turbine can provide greater inertial energy, in the process of transient DC fault, in order to keep the capacitor voltage constant, its current inner loop given value and feedback value fluctuate greatly, resulting in overcurrent blocking of the converter. Fault ride-through failure. In the method proposed in this paper, the VSG outer loop is constructed based on the capacitor energy storage of the power module. During the transient fault process, the given value of the power module voltage will be increased, which is the same as the change trend of the feedback value, thereby reducing the fluctuation of the current inner loop.

(3) Please find the gap first in the introduction section by discussing several recent studies in VSG. Then, write your contributions.

In the first part, the gap of recent studies and our contributions of the paper are explicitly presented, supplemented as follows: 

“For example, the inertial energy in the aforementioned VSG method needs to be provided by an AC power grid, an energy storage system or a wind turbine, which requires an additional device or system to provide or consume the inertial energy. The power module capacitor of the MMC itself can be used to store energy and provide inertia. For this aspect, the main contributions of this paper include: proposing an inertial simulation method for MMC topology, designing the current inner loop to improve the transient performance, and analyzing the impedance characteristics of the converter under the proposed control strategy. The converter under the proposed control scheme can provide transient inertia for the AC system, improve the transient fault ride-through capability of the system, and provide an optimized control scheme for the weak AC system to connect to the large-capacity power electronic converter.” 

(4) "For VSG control structure and inertial equivalent method, there is still room for optimization." Is there any optimization the proposed method?

The VSG outer loop based on power module capacitor energy storage is proposed, which utilizes the MMC's own capacitance to provide a certain inertia without relying on additional energy storage. The following content is added in the section 1.

The main contributions of this paper include: proposing an inertial simulation method for MMC topology, designing the current inner loop to improve the transient performance, and analyzing the impedance characteristics of the converter under the proposed control strategy. The converter under the proposed control scheme can provide transient inertia for the AC system, improve the transient fault ride-through capability of the system, and provide an optimized control scheme for the weak AC system to connect to the large-capacity power electronic converter.

(5) Why dq frame control strategy is used?

The current inner loop of the converter can realize the decoupling of active and reactive power in the dq frame, so the dq frame is still used in the VSG control.

(6) How the controller parameters have been tuned? If the tuning has been done based on the frequency response analysis, a complete small-signal model is required?

The converter adopts a double closed-loop control structure. The VSG control proposed in this paper is an outer loop. Through the derivation of impedance characteristics and the results of impedance scanning (as shown in Fig. 5), VSG mainly affects the low-frequency characteristics of the system, so the small signal model is mainly designed for the outer loop of VSG. About the controller parameters, the supplementary explanation in section 4 is as follows.

The control parameters of the VSG outer loop are determined by the impedance characteristics of the small signal model, and the current inner loop controller parameters are mainly adjusted through simulation, so that the adjustment speed of the current inner loop is significantly higher than that of the outer loop.

(7) How the capacitor voltage balancing is achieved during worst case scenario?

Voltage sequencing method is adopted for keeping power module voltage balancing. Supplementary explanation and references in section 4 are as follows.

“And the power module voltage sequencing method is used to maintain voltage balance during both steady state and transient state [25].”

(8) The quality of the figures of the simulations results should be improved.

All the figures of the simulations results Fig.9-11 have been redrawn and they will remain sharp when zoomed in.

(9) Is the proposed control system a multi input multi output (MIMO)? Please explain.

It is a MIMO system (2 inputs-2 outputs), but it can be decoupled in the dq frame, so more complicated MIMO analysis method is not used. Only for the VSG outer loop proposed in this paper, the impedance characteristics between the input and the output of the VSG outer loop are analyzed. The application of MIMO method will be taken as one of the future research works.

---

## [Decision Letter · Decision Letter 1]

8 Aug 2022

PONE-D-22-05125R1Inertial Imitation Method of MMC with Hybrid Topology for VSC-HVDCPLOS ONE

Dear Dr. YIN,

Thank you for submitting your manuscript to PLOS ONE. After careful consideration, we feel that it has merit but does not fully meet PLOS ONE’s publication criteria as it currently stands. Therefore, we invite you to submit a revised version of the manuscript that addresses the points raised during the review process.

We look forward to receiving your revised manuscript.

Kind regards,

Yogendra Arya

Academic Editor

PLOS ONE

Journal Requirements:

Reviewers' comments:

Reviewer's Responses to Questions

**Comments to the Author**

1. If the authors have adequately addressed your comments raised in a previous round of review and you feel that this manuscript is now acceptable for publication, you may indicate that here to bypass the “Comments to the Author” section, enter your conflict of interest statement in the “Confidential to Editor” section, and submit your "Accept" recommendation.

Reviewer #1: All comments have been addressed

Reviewer #2: All comments have been addressed

2. Is the manuscript technically sound, and do the data support the conclusions?

Reviewer #1: Yes

Reviewer #2: Yes

3. Has the statistical analysis been performed appropriately and rigorously? 

Reviewer #1: Yes

Reviewer #2: No

4. Have the authors made all data underlying the findings in their manuscript fully available?

Reviewer #1: Yes

Reviewer #2: No

5. Is the manuscript presented in an intelligible fashion and written in standard English?

Reviewer #1: Yes

Reviewer #2: Yes

6. Review Comments to the Author

Reviewer #1: This is a revised manuscript, and is suitable for acceptance and publish now. The methods and results of the manuscript are helpful for researchers and VSC-HVDC engineering application.

Reviewer #2: Please consider the following comments.

1. No numerical results have been added in the abstract. The authors just mentioned "The results show...."

2. ["For VSG control structure and inertial equivalent method, there is still room for optimization." Is there any optimization the proposed method?] this comments have not been addressed well

7. PLOS authors have the option to publish the peer review history of their article (what does this mean?). If published, this will include your full peer review and any attached files.

Reviewer #1: No

Reviewer #2: No

---

## [Author Response · Author response to Decision Letter 1]

14 Sep 2022

Thank you for allowing a revised version of our manuscript, with an opportunity to address the reviewers’ comments.

Reviewer #1:

There is no question.

Reviewer #2:

1. No numerical results have been added in the abstract. The authors just mentioned "The results show...."

The results were added in abstract as “The simulation results show that it can effectively improve the current control capability during the transient process for systems with a 1:2 ratio of converter capacity to grid capacity (The grid short-circuit capacity is 60MW and the MMC is 30 MW).” and “The results show that the power adjustment time of MMC under the proposed VSG control is about 1s, while the adjustment time under the conventional control strategy is greater than 4s.”

2. "For VSG control structure and inertial equivalent method, there is still room for optimization." Is there any optimization the proposed method?

The optimizable aspects are described separately from two aspects, which are modified and supplemented in the section 1 as follows: 

For VSG control structure and inertial equivalent method, there is still room for optimization. For control structures, the modulation signal is directly output by the traditional VSG controller, without direct control of the AC current, and the fault ride-through capability is poor. Therefore, a suitable current inner loop can be constructed to improve the transient characteristics of the system. For inertial equivalent, the inertial energy in the aforementioned VSG method needs to be provided by an AC power grid, an energy storage system or a wind turbine, which requires an additional device or system to provide or consume the inertial energy. The power module capacitor of the MMC itself can be used to store energy and provide inertia.

---

## [Decision Letter · Decision Letter 2]

31 Oct 2022

Inertial Imitation Method of MMC with Hybrid Topology for VSC-HVDC

PONE-D-22-05125R2

Dear Dr. YIN,

We’re pleased to inform you that your manuscript has been judged scientifically suitable for publication and will be formally accepted for publication once it meets all outstanding technical requirements.

Kind regards,

Yogendra Arya

Academic Editor

PLOS ONE

Additional Editor Comments (optional):

Reviewers' comments:

Reviewer's Responses to Questions

**Comments to the Author**

1. If the authors have adequately addressed your comments raised in a previous round of review and you feel that this manuscript is now acceptable for publication, you may indicate that here to bypass the “Comments to the Author” section, enter your conflict of interest statement in the “Confidential to Editor” section, and submit your "Accept" recommendation.

Reviewer #1: All comments have been addressed

Reviewer #2: All comments have been addressed

2. Is the manuscript technically sound, and do the data support the conclusions?

Reviewer #1: Yes

Reviewer #2: Yes

3. Has the statistical analysis been performed appropriately and rigorously? 

Reviewer #1: Yes

Reviewer #2: N/A

4. Have the authors made all data underlying the findings in their manuscript fully available?

Reviewer #1: Yes

Reviewer #2: Yes

5. Is the manuscript presented in an intelligible fashion and written in standard English?

Reviewer #1: Yes

Reviewer #2: Yes

6. Review Comments to the Author

Reviewer #1: This is the revised manuscript. The authors have made positive responce to reviewers' comment to meet the requirement for publishing in the Journal.

Reviewer #2: All comments have been addressed.

All comments have been addressed.

All comments have been addressed.

7. PLOS authors have the option to publish the peer review history of their article (what does this mean?). If published, this will include your full peer review and any attached files.

Reviewer #1: No

Reviewer #2: No

---

## [Editor Report · Acceptance letter]

1 Dec 2022

PONE-D-22-05125R2 

Inertial Imitation Method of MMC with Hybrid Topology for VSC-HVDC 

Dear Dr. YIN:

I'm pleased to inform you that your manuscript has been deemed suitable for publication in PLOS ONE. Congratulations! Your manuscript is now with our production department. 

Kind regards, 

on behalf of

Dr. Yogendra Arya 

Academic Editor

PLOS ONE